# Study of Brewer’s Spent Grain Environmentally Friendly Processing Ways

**DOI:** 10.3390/molecules28114553

**Published:** 2023-06-05

**Authors:** Konstantin V. Kobelev, Irina N. Gribkova, Larisa N. Kharlamova, Armen V. Danilyan, Maxim A. Zakharov, Irina V. Lazareva, Valery I. Kozlov, Olga A. Borisenko

**Affiliations:** All-Russian Scientific Research Institute of Brewing, Beverage and Wine Industry—Branch of V.M. Gorbatov Federal Research Center for Food Systems, 119021 Moscow, Russia; k.kobelev55@mail.ru (K.V.K.); harlara@yandex.ru (L.N.K.); a.danilyan@mail.ru (A.V.D.); mazakharoff@mail.ru (M.A.Z.); lazirka@gmail.com (I.V.L.); izhlineo@yandex.ru (V.I.K.); smbiologic@rambler.ru (O.A.B.)

**Keywords:** brewer’s spent grain, processing, organic compounds, brewer’s spent grain structure, ECA treatment water

## Abstract

Background: This article is devoted to the study of the effect of electrochemically activated water (catholyte with pH 9.3) on organic compounds of the plant matrix of brewer’s spent grain in order to extract various compounds from it. Methods: Brewer’s spent grain was obtained from barley malt at a pilot plant by mashing the malt followed by filtration and washing of the grain in water and storing it at (0 ± 2) °C in craft bags. For the organic compound quantitative determination, instrumental methods of analysis (HPLC) were used, and the results were subjected to mathematical analysis. Results: The study results showed that at atmospheric pressure, the alkaline properties of the catholyte showed better results compared to aqueous extraction with respect to β-glucan, sugars, nitrogenous and phenolic compounds, and 120 min was the best period for extraction at 50 °C. The excess pressure conditions used (0.5 ÷ 1 atm) revealed an increase in the accumulation of non-starch polysaccharide and nitrogenous compounds, while the level of sugars, furan and phenolic compounds decreased with increasing treatment duration. The waste grain extract ultrasonic treatment used revealed the effectiveness of catholyte in relation to the extraction of β-glucan and nitrogenous fractions; however, sugars and phenolic compounds did not significantly accumulate. The correlation method made it possible to reveal the regularities in the formation of furan compounds under the conditions of extraction with the catholyte: Syringic acid had the greatest effect on the formation of 5-OH-methylfurfural at atmospheric pressure and 50 °C and vanillic acid under conditions of excess pressure. Regarding furfural and 5-methylfurfural, amino acids had a direct effect at excess pressure. It was shown that the content of all furan compounds depends on amino acids with thiol groups and gallic acid; the formation of 5-hydroxymethylfurfural and 5-methylfurfural is influenced by gallic and vanillic acids; the release of furfural and 5-methylfurfural is determined by amino acids and gallic acid; excess pressure conditions promote the formation of furan compounds under the action of gallic and lilac acids. Conclusions: This study showed that a catholyte allows for efficient extraction of carbohydrate, nitrogenous and monophenolic compounds under pressure conditions, while flavonoids require a reduction in extraction time under pressure conditions.

## 1. Introduction

Brewer’s spent grain is a brewing industry byproduct that is the grain shell’s layers [1]. Cuomo et al. have recently [2] released data on the wide uses of brewer’s spent grain in different branches of industries. This is due to the rich organic composition of the processed grain residue plant matrix, which is characterized by the connections of the complex substance. Brewer’s spent grain, making up 85% of brewing waste, is reported to be rich in protein, dietary fiber (insoluble—cellulose, arabinoxylan, lignin, as well as soluble—β-glucan, starch, pectin compounds) and phenolic compounds [1,3,4].

It should be noted that brewer’s spent grain organic structure processing is of a directed nature, and its choice depends on a few criteria, such as the fields in which the spent grain products are used, the environmental performance of the techniques applied, the economic feasibility, a high yield and the minimization of byproducts in the end product of the processing.

The impact of the physical method on brewer’s spent grain structure, such as temperature, pressure, ultrasound, microwaves, electric fields, etc. [5,6,7,8], with acid–base treatment [1] is eco-friendly.

However, alkaline and acidic reagent application does not always comply with the aim of eco-friendly brewer’s spent grain processing.

In this case, in various industries, including the food industry, it is considered appropriate to use electrochemically activated water (ECA-water) solutions (anolyte and catholyte), possessing acidic (pH of 3.0–3.5) and alkaline (pH of 9.0–9.5) properties [9,10,11,12] due to the change in the number of positively charged H+ ions or negatively charged OH^−^ ions with an applied electric field [13,14,15].

The water structure chemical activation mechanism [16] is based on a constant voltage between the electrodes’ passing, which results in water electrolysis. Oxygen is produced at the anode, and the water has disinfecting, acidic properties during the reaction:2H_2_O → O_2_ + 4H^+^ + 4e^−^

Hydrogen is produced at the cathode, and the water obtains a basic pH:2H_2_O + 2e^−^ → H_2_^+^ + 2OH^−^

Table 1 shows the application of electrochemically activated water.

This study aimed to investigate how brewer’s spent grain structure is transformed under the impact of physical factors (temperature, pressure, ultrasound) in the presence of an ECA-water catholyte solution.

## 2. Results

This study’s model was structured in such a way so that the effect of the factors, applicable to the deep processing of the brewer’s spent grain plant matrix, could be understood through comparison. In the first stage, the impact of atmospheric pressure, a temperature of 50 °C and a catholyte with a pH of 9.3 were studied. In the second stage, excess pressure and the catholyte effect were studied. In addition, in the third stage, the conditions of ultrasound (40 kHz) and the catholyte impact on the structure of the brewer’s spent grain organic compounds were studied.

### 2.1. Study of ECA-Water Catholyte Influence on Brewer’s Spent Grain Extract Composition under Atmospheric Pressure Conditions

Table 2 shows the brewer’s spent grain extract organic compounds obtained at a temperature of 50 °C and under atmospheric pressure.

The results of Table 2 show that under atmospheric pressure, β-glucan extraction is affected by the extraction duration and the solvent nature. Thus, an increase in the extraction time treatment by 1 h contributes to an increase of non-starch polysaccharide content in the extract by 20% in the control sample (when aqueous extraction is applied) due to liquid diffusion into the brewer’s spent grain structure and by 40% if a catholyte is used as a solvent of basic nature.

When brewer’s spent grain is being processed for 1 h using a catholyte, the β-glucan content in the extracts increases by 25% and when the treatment lasts for 2 h, by 45.7% compared to aqueous extraction under the same conditions.

It should be noted that previously, the non-starch polysaccharide was successfully extracted via an enzymatic breakdown of starch and proteins by a complex of α-amylase and protease when using alkaline extraction of arabinoxylans and (1-3,1-4)-β-d-glucan [19]. NaOH in combination with NaBH_4_ was used as a basic reagent to limit hydrolysis, which resulted in an increase in the hemicellulose breakdown of up to 20% as well as up to 1.8% of the brewer’s spent grain weight of (1-3,1-4)-β-d-glucan. According to other studies [20], (1-3,1-4)-β-d-glucan extraction was carried out under conditions of exposure to microwave radiation. The undissolved starch, associated with (1-3,1-4)-β-d-glucan, was previously removed at a temperature of 140 °C. This allowed for the achievement of a (1-3,1-4)-β-d-glucan yield of 0.4 mol.%.

Application of solvent extraction, ultrasound-assisted extraction, microwave-assisted extraction or reverse extraction facilitates a high yield of (1-3,1-4)-β-d-glucan up to 8.8%, 0.3%, 0.3% and 2.2%, respectively. The use of aqueous extraction under pressure (up to 50 bar) at a temperature varied in the range (110 ÷ 180) °C when the process lasts for 15–45 min allows for an extraction yield of (1-3,1-4)-β-d-glucan with an average molecular mass of 160 kDa at a level up to 54.0% [21], while extraction carried out at 55 °C and under atmospheric pressure within 3 h leads to lower yields of 53.7% compared with 73.2% of β-glucan with a molecular mass of 200 kDa compared with 55 kDa under excess pressure [22]. In our case, the yield of β-glucan was 0.35% when the process lasted for 1 h and 0.50% under 2 h extraction using a catholyte.

Monosaccharides (xylose, glucose, fructose) can be extracted in the presence of a catholyte within 1 h. Under other conditions, they reduce their trace amount or completely disappear, which may be due to the formation of oxidation products (furan compounds) taking place during hydrolysis [23]. Of course, an acid’s presence in the medium is important for obtaining furan compounds. Here, we can talk about acetic acid, which is produced because of the autohydrolysis process involving OH^−^ ions of catholyte and arabinoxylan molecules [24]. The furan compound’s content depends only on the time during which the organic material is processed since the control and experimental values are within the permissible errors.

The soluble nitrogen level extracted from the brewer’s spent grain structure remains unchanged regarding the temperature and the catholyte presence in the samples and is at a level of 117.7 ÷ 126.1 mg/L since the protein compound content is within the measurement method error. The amino nitrogen (FAN) concentration increases depending on the time of catholyte treatment: There is an increase in the FAN content by 16% in the samples obtained within 120 min compared to a relatively shorter process (60 min). As to the catholyte effect, it allows for an increase in the level of FAN extraction of 14% within 60 min and 20.8% within 120 min.

Table 2 shows the characteristics of low molecular mass fraction FAN and soluble nitrogen. These characteristics differ because the low molecular weight fraction’s soluble nitrogen comprises non-protein nitrogen. The catholyte effect allows for an increase in the low molecular weight fraction’s soluble nitrogen, obtaining 23% within 60 min and 41.6% within 120 min compared to the control. This means that both the treatment period and catholyte presence have an effect. It should be noted that the non-protein nitrogen content is increased only in the presence of the catholyte. If we judge by the difference between the low molecular weight fraction’s soluble nitrogen and FAN, it is shown that the catholyte enables the extraction of non-protein nitrogen from the brewer’s spent grain matrix in the range of 5.6 ÷ 6.0 mg/L. The process duration does not significantly affect the extraction process.

Nitrogen with thiol groups characterizes the presence of amino acids with SH groups in a brewer’s spent grain matrix, which are the products of hordein hydrolysis [25]. According to other authors, acid–base treatment application [26] obtains 72% of protein nitrogen with a mass of more than 10 kDa, i.e., peptones, and 21% of nitrogen with a molecular weight less than 5 kDa, i.e., the nitrogen of amino acids. The results we obtained show (Table 2) that amino acids with thiol groups decrease their content during the longer extraction process (120 min). Thus, with 60 min and 120 min extraction periods under catholyte basic conditions, it is possible to extract nitrogen with thiol groups 90% and 45%, respectively, more than compared to the control. It should be noted that the amino acids with the thiol group’s contribution decrease equally in the case of the catholyte application when the process lasts for 1 or 2 h. However, a more objective assessment of the SH-amino acids to the total amino acid’s amount ratio suggests that considering all the conditions for amino nitrogen transformation, the SH-amino acid contribution in the case of using a catholyte is higher than the control values in all cases.

Brewer’s spent grain is a source for the extraction of phenolic compounds [27].

The sources of low molecular weight phenolic compounds in the structure of brewer’s spent grain are lignin [28], ferulic and p-coumaric acids, which account for up to 70% of the total phenol content, as well as sinapic and caffeic acids. Ferulic acid is known to be associated with proteins, lignin, hemicellulose and/or polysaccharides via etheric bonds [28,29]. It is noted that the temperature affects a monophenolic compound’s form, and they can undergo a decarboxylation reaction to form vinyl derivatives of phenols [27].

There is extensive data in the literature providing information on the content of phenolic compounds in extracts. Thus, Ladecola et al. [30] report at 80 °C within 50 min in an extract of 65:35% ethanol when subjected to a 37 kHz ultrasound treatment, the presence of ferulic acid 1.5 mg/L, vanillic acid 0.78 mg/L and p-coumaric acid 0.12 mg/L. According to Table 2, the vanillic acid content did not change depending on the treatment time and the solvent type applied and was within the range of 0.08 ÷ 0.098 mg/L; the syringic acid content depended on the extraction period. Within 60 min, the concentration reached 0.06 ÷ 0.07 mg/L and within 120 min, 0.107 ÷ 0.11 mg/L, respectively. The vanillin content became 2.3 times higher within 60 min of catholyte treatment compared to the control, and within 120 min of the extraction treatment using water and a catholyte at 50 °C, it was undetectable. The syringaldehyde content varied significantly depending on the exposure duration: During aqueous extraction, within 120 min, 10% more of the aldehyde was released compared to a less prolonged extraction period. The mild alkaline catholyte conditions did not significantly affect syringaldehyde extraction. Apparently, more aggressive methods of basic treatment are required, which is confirmed by the authors [31].

Petron et al. [32] declare that gallic acid and catechin are present in aqueous extracts of brewer’s spent grain. Indeed, gallic acid was extracted regardless of the treatment time and solvent type, and its concentration was 1.62 ÷ 1.75 mg/L (Table 2).

Regarding catechin extraction, it was found to be dependent on the treatment time during extraction using water. When the treatment time was 2 times longer, the catechin content also went up by 25%. The catholyte also has an effect. Regardless of the extraction duration, basic catholyte conditions allowed for an increase of the catechin yield by 26.5% on average (Table 2).

The brewer’s spent grain extracted phenolic compounds also comprise rutin and quercetin [33]. According to Table 2, the rutin and quercetin content increased by 2.3 times and by 36%, respectively, when treated with the catholyte for 120 min compared to a 60 min treatment time.

### 2.2. Study of the ECA-Water Catholyte Effect on Brewer’s Spent Grain Extract Composition under Excess Pressure Conditions

Table 3 shows the brewer’s spent grain extract organic compounds obtained under conditions of excess pressure.

The results presented in Table 3 represent the effect of a number of factors on organic compound extraction. With regard to β-glucan extraction, it should be noted that an increase in excess pressure from 0.5 to 1 atm reduces the non-starch polysaccharide water extraction efficiency by 13–18% within 30–120 min of treatment. The longer the extraction lasts, the lower the decrease in concentration is. The use of excess pressure has been studied previously in relation to β-glucan extraction in yeast [34]. β-glucan water extraction conditions at a temperature of 121 °C and under an excess pressure of 1.1 atm within 1–5 h facilitate its extraction from the extraction object structural polyoses [35]. In our opinion, the effect of the extraction conditions within the period of 120 min and under a pressure of 1 atm can cause the destruction of the units of the β-glucan glucose chain, which affects its detection conditions.

With regard to brewer’s spent grain structure processing via excess pressure and a catholyte, it can be noted that compared to the non-starch saccharide concentration obtained with 30 min of processing and under pressure of 1 atm, the analog content obtained under the same conditions, but within 60 and 120 min of treatment, increases by 38% and 52%, respectively.

Regarding monosaccharides, it should be noted that a pressure of 1 atm, catholyte presence and a 30 min extraction treatment period allow for accumulating insignificant xylose, glucose and fructose content in the medium (Table 3). Under extraction conditions for a duration of 60 min, sugar accumulation occurs uniformly in all samples (aqueous and catholyte extracts), whereas under a pressure of 1 atm with a 120 min treatment, xylose and glucose are released more intensively in the presence of the catholyte, but they thermally decompose or enter a reaction of melanoidin formation at the high temperature of the process (120 °C). This leads to a decrease in the sugar concentration [36].

Fructose, with 120 min of treatment under a pressure of 0.5 atm and 1 atm, is accumulated 30% and 50% more intensively in the presence of a catholyte.

Regarding furan compounds, it is known that the sources of 5-hydroxymethylfurfural and 5-methylfurfural are glucose and fructose, respectively, which are facilitated by elevated temperatures and the Maillard reaction [37], and of furfural is five-carbon sugars [38]. Furan compound accumulation indicates the intensity at which free sugars enter the melanoidin formation reaction. As shown in Table 3, a 110 °C (0.5 atm) temperature of the medium and water extraction within 60 and 120 min can make the glucose conversion rate into 5-hydroxymethylfurfural 5 and 30 times higher, respectively, compared to a 30 min extraction, and a 120 °C (1 atm) medium temperature is 4 and 6.4 times higher, respectively. Thus, it is more likely that at 120 °C, other sugar decomposition products, which are uncontrolled in this study, such as formic, lactic and levulinic acids, can be formed [39].

Under catholyte conditions, a 110 °C (0.5 atm) temperature of the medium and extraction within 60 and 120 min can make the glucose conversion rate into 5-hydroxymethylfurfural 54% and 10 times higher, respectively, compared to a 30 min extraction, and the 120 °C (1 atm) medium temperature is 4 and 10 times higher, respectively.

With regard to the 5-methylfurfural transformation in aqueous extracts obtained at a temperature of 110 °C (0.5 atm), the highest concentration has solutions obtained within 60 min. The 5-methylfurfural content is 7 times higher than that in the 30 min extract analog. In aqueous extracts obtained at 120 °C (1 atm), the most effective extraction duration is also 60 min. In this case, the 5-methylfurfural content is 2.6 times higher than that in the 30 min extract analog.

Additionally, 5-methylfurfural extraction using a catholyte proved to be effective under a pressure of 0.5 atm (110 °C) within a 60 min exposure period. In other cases, this furan compound was not detected. Thus, catholyte conditions promote reactions of deep sugar decomposition [39]. The same is true with respect to pentose decomposition and furfural formation.

As shown in Table 3, the duration process is very important when water extraction is applied. Thus, the soluble nitrogen level has a maximum value when the extraction period is 60 min, and under a pressure of 0.5 atm and 1 atm, there is an increase of 53% and 24%, respectively, compared to a 30 min period of extraction treatment. Catholyte use as a solvent substance makes it possible to achieve the maximum soluble nitrogen yield within 120 min under a pressure of 0.5 atm (an increase of 59% compared to the yield obtained with 30 min of extraction) and within 60–120 min under 1 atm of pressure (an increase of 29% compared to the yield obtained with 30 min of extraction). The obtained data showed a decrease of 7–15% in the soluble nitrogen level of the aqueous extracts under pressures of 0.5 atm and 1 atm with an extraction duration of 120 min compared to the maximum value. The same applies to the catholyte extracts under a pressure of 1 atm with a duration of extraction of 120 min (by 10% if compared with the maximum value). The authors state that the abrupt hydrolysis of nitrogen compounds is observed at 190 °C [40]. It is a more complex process compared to carbohydrates; thus, it is possible to conclude that the losses are related to the Maillard reaction.

The FAN content varies under excess pressure and 110–120 °C within 30 ÷ 120 min during aqueous and catholyte extraction. There is a general trend characteristic of both the experimental and control samples treated within 30 min regardless of the pressure and temperature. The control value is characterized by a range of 58.8 ÷ 61.6 mg/L, and the experimental value is characterized by a range of 72.8 ÷ 75.6 mg/L, which is actually within the detection method error range. However, there is an increase in the FAN content by 24% compared to the control value due to the alkaline medium. Under neutral pH conditions, the FAN content is accumulated only under a pressure of 1 atm and a treatment of 60 min, while the treatment of 120 min under a pressure of 0.5 atm, which is, 110 °C, leads to the FAN level of the aqueous extract obtained under a pressure of 1 atm but within 60 min. The exposure under a pressure of 1 atm within 120 min helps reduce the FAN level due to their inclusion in the Maillard reaction. Similarly, when the pH of the solvent is 9.3, more amino acids are spent (by 20%) only within 120 min under a pressure of 1 atm.

Under alkaline conditions, besides the Maillard reaction, chemical transformations (racemization) can occur in which hydrogen breaks down from the carbon atom at a position α, and a planar carbanion structure is formed [41]. The racemization depth depends on the free or bound amino acid form, temperature and medium pH, as well as on the nature of the amino acid R group. At the same time, the authors state [42] that the dipeptide amino acid N-terminal was subjected to isomerization to a greater extent than the C-terminal residue.

The non-protein nitrogen value, characterizing the brewer’s spent grain organic polyose cleavage hydrolytic processes depth, was observed under excess pressure during treatment with an aqueous solvent for 30 min, and the higher the pressure/temperature, the higher the yield. When the treatment duration increased, the non-protein nitrogen either reduced its content or was beyond the detection scope. As for the hydrolytic conditions under a pH of 9.3 (catholyte treatment), the non-protein nitrogen halved its content (0.5 atm, 60 min) or was outside the detection limits.

Regarding the SH-amino acids, according to the data in Table 3, it can be said that the excess pressure value and the process duration impacted the amino acid extraction/conversion degree, which affected their final concentration in the aqueous extracts. Thus, when the extraction time was the shortest (30 min), the maximum SH-amino acid amount was accumulated—23.7 µM/L under a pressure of 0.5 atm and 31.6 µM/L under a pressure of 1 atm, which indicates a greater extraction force at 120 °C. When the extraction time increased (60–120 min), the level of the accumulated amino acids decreased by 3 and 3.5 times within 60 and 120 min of extraction. The use of the catholyte in the processing of the brewer’s spent grain structure under a pressure of 0.5 atm promoted the reduction of the extracted SH-amino acid amount to 19.2 ÷ 27.4 µM/L and under a pressure of 1 atm, to 15.8 ÷ 26.3 µM/L. Moreover, the longer the processing period, the greater the amount of amino acids that were extracted by means of the catholyte under a pressure of 0.5 atm and a lesser amount under a pressure of 1 atm.

When analyzing the SH-amino acid amount to the amino acid total number ratio, it is possible to say that the maximum amount of SH-nitrogen compounds accumulated over the period of 60 to 120 min during aqueous extraction under excess pressure of 0.5 ÷ 1 atm.

The vanillic acid concentration during aqueous extraction was within the range of 0.09 ÷ 0.29 mg/L, and during catholyte extraction, it was 0.09 ÷ 0.20 mg/L under a pressure ranging from 0.5 atm to 1 atm. When extracted under neutral pH extraction conditions, the vanillic acid concentration increased with a 120 min extraction period. Under a pH of 9.3, the extraction values within the 60 ÷ 120 min extraction period were 0.13 ÷ 0.20 mg/L, which is within the range of the method detection error.

The syringic acid content during aqueous extraction under excess pressure reached the accumulation maximum within 120 min and is 0.17 ÷ 0.19 mg/L, whereas under pH of 9.3 conditions, only under a pressure of 0.5 atm did the phenolic compound accumulation reach a level of 0.13 ÷ 0.15 mg/L. Excess pressure of 1 atm/120 °C almost halves the syringaldehyde concentration under a pH of 9.3.

Regarding the other phenolic compounds (syringaldehyde, sinapaldehyde and vanillin), we should say that catholyte use and an excess pressure of 0.5–1 atm allowed us to achieve maximum extraction values within 120 min.

The catechin highest content extracted under an excess pressure of 1 atm within 60 min was 7.43 mg/L, which is 50% higher compared to the control. In the rest of the cases, aqueous extraction proved to be more effective than the catholyte one. The rutin extraction using the catholyte under a pressure of 1 atm within 30 min was 4 times higher compared to the control. In all other cases, rutin was below the control values. The same effect was observed during quercetin extraction. The best extraction conditions were the following: 1 atm, 30 min and the presence of the catholyte. This allowed us to achieve a quercetin recovery level that is 2.3 times higher than that in aqueous extracts.

### 2.3. Study of ECA-Water Catholyte Influence on Brewer’s Spent Grain Extract Composition under Atmospheric Pressure and Ultrasound Conditions

Table 4 shows the brewer’s spent grain extract organic compounds obtained under atmospheric pressure and ultrasound conditions.

The results presented in Table 4 show, with respect to β-glucan extraction efficiency, that the prolonged (60 ÷ 120 min) extraction process under ultrasound conditions is the most effective. This allows for increasing the efficiency of extraction by 14% compared to aqueous extraction carried out under the same conditions. Pentoses (arabinose and xylose) have a maximum yield during the 2 h extraction process under cavitation conditions, which occur when the catholyte is subjected to ultrasound [43]. For hexose (glucose and fructose), only ultrasonic extraction in the aqueous solution within 30 min is effective. There are data in the literature confirming the destructive ultrasound effect on polysaccharide structure [44,45]. The authors found that at frequencies of 20/40 and 20/60 kHz, the conformation of the triple spiral of the polysaccharide into a loose, flexible triple spiral takes place.

The authors [46] claim that the aqueous extracts processing for a shorter duration were found to be more effective for the efficient isolation of monosaccharides.

Furan compounds were detected only when the time of extraction was 30 min both in aqueous and catholyte extracts. They were detected in trace amounts and did not significantly differ from each other.

When subjected to ultrasound, nitrogen compounds were extracted in greater amounts when the ultrasound treatment duration was 60–120 min, which was higher compared to the control values in aqueous extracts by 13–16%. FAN was recovered more efficiently in the catholyte medium when exposed to ultrasound within 30 min; the FAN concentration was 18% higher than in aqueous extracts (Table 4).

Since the low molecular weight fraction’s soluble nitrogen was quantitatively smaller than the amino nitrogen, it was not possible to calculate the amount of non-protein nitrogen extracted by means of ultrasound.

The nitrogen with the thiol group’s concentration was 6 times higher compared to the control value under the ultrasonic condition extraction in the catholyte presence when the extraction duration was 60 min. When the extraction duration was 30 min and 120 min, the nitrogen substance’s group experimental values were 80% and 2.3 times higher compared to the control, correspondingly. This effect is explained by the sonomechanical modification of nitrogen structures at the level of protein chains or the structures of peptides or amino acids, which are the proteins’ structure parts. This can result in temporary or stable nitrogen compound structure transformation [47]. Yao et al. confirmed a limited increase in thiol extraction when subjected to ultrasound [48]. Considering the ratio of thiol nitrogen and the total number of amino acids, the most effective conditions were the 30 min process duration in the presence of the catholyte. However, it was noted that the aqueous extraction while using the ultrasonic treatment was always more effective than the catholyte one.

Of all phenolic monomeric compounds, only syringaldehyde was extracted. The effective conditions for its recovery were catholyte extraction within 30 min, which allowed for an increase in its content in the extract by 48% compared to aqueous extraction (Table 4).

Catechin decreased its concentration in the presence of the catholyte by 2–4 times compared to aqueous extraction, which is due to its oxidation under the impact of active oxygen formed as a result of cavitation in the medium [49]. Rutin was also better extracted in the aqueous medium within 30 min, while catholyte extraction contributed to a rutin concentration loss of 50% during extraction treatment for 30 min; longer extraction treatment caused greater losses. Quercetin was 41% better extracted under catholyte conditions during extraction treatment within 30–60 min compared to aqueous extraction.

### 2.4. Study of the Influence of Brewer’s Spent Grain Extract Organic Compounds on Furan Compound Formation

The conditions under which furan compounds are accumulated as well as the impact of other organic compounds on their formation when a catholyte is used are not fully clear. With this purpose, the principles of multifactorial correlation–regression correlation were applied.

Table 5 and Table 6 reflect the main parameters characterizing the bond strength with the process of furan compound formation when the spent grain structure is catholyte-treated.

Considering the value of the furan compound’s paired correlation coefficients (Table 5), it is possible to say that syringic acid has the greatest effect on the formation of 5-OH-methylfurfural (atmospheric pressure, 50 °C). A similar effect has vanillic acid under excess pressure.

Regarding furfural and 5-methylfurfural, amino acids have a direct effect under excess pressure.

The data in Table 6 are grouped as follows: All furan compound content depends on amino acids with thiol groups and gallic acid; the formation of 5-OH-methylfurfural and 5-methylfurfural is impacted by gallic and vanillic acids; the isolation of furfural and 5-methylfurfural is determined by amino acids and gallic acid; excess pressure conditions facilitate furan compound formation under gallic and syringic acid’s influence. The general effect of free amino acids on furan compound formation under conditions of atmospheric and excess pressure is observed.

Since the furan compound content was beyond detection limits regarding the conditions under which the brewer’s spent grain plant matrix was treated, it was not possible to obtain statistical dependencies when subjected to ultrasound.

## 3. Discussion

Summarizing the data obtained (Table 2, Table 3 and Table 4), the most effective conditions for the extraction and concentration of the substances are presented in Table 7.

The data in Table 7 show that given the mild effect of the catholyte with a pH of 9.3, based on the interaction of the OH^−^ ion and organic compound groups, excess pressure was the most effective way to treat the brewer’s spent grain structure regarding carbohydrates, most nitrogen compounds and quercetin. Nitrogen with thiol groups, catechin and rutin appeared to be the most sensitive compounds, which are effectively extracted under atmospheric pressure and 50 °C.

The organic compounds’ recovery possibility is dependent on their position, bonds with other compounds of the plant matrix and their properties.

β-glucan is mainly occurring in the aleurone, subaleurone and endosperm of some cereals; therefore, it is present in the structure of the cereal grain outer layers, representing the brewer’s spent grain. The β-glucan molecular structure has a high ability to bind water, which determines its physicochemical properties, such as solubility, viscosity and gelation [50]. Therefore, the electrochemically charged water use facilitates the extraction of this non–starch polysaccharide, and the efficiency of its recovery correlates with the temperature of the medium; the higher the temperature, the greater the rate of the extract penetration into the grain subaleyron structure where β-glucan is located. It has also been shown that the physical method’s application for β-glucan isolation promotes its extraction without affecting the structure of other carbohydrate compounds (for example, arabinoxylan) and provides a yield of 0.4 mol.% of (1-3,1-4)-β-d-glucan [21]. In our case, the yield of physically treated β-glucan was 0.609%, which was the most significant value of its yield of all the applied physical treatment methods using the catholyte.

Monosaccharides are the products of carbohydrate series polyose hydrolysis (arabinoxylans, cellulose, β-glucan, pectin compounds) within the brewer’s spent grain structure. Therefore, their number in extracts is proportional to the depth of impact on the rigid structures of the brewer’s spent grain. Xylan molecules are the main component of hemicellulose, and brewer’s spent grain hydrothermal treatment may result in the formation of various structure derivatives (from monosaccharides to various substituted xylooligosaccharides) [51]. The brewer’s spent grain treatment extreme conditions using water are known (150 °C), which facilitated an arabinose yield in the amount of 5%, xylose—1% in a mixture with polymers, compounding 42% of arabinose, xylose and glucose with a total content of 18, 56 and 16 mol.%, respectively [52]. It should be noted that an increase in temperature releases more arabinose compared to xylose due to its higher thermal sensitivity [36]. This fact is in agreement with the results that were obtained (Table 7) when the extraction conditions (0.5 atm or 110 °C) were more effective for xylose recovery. According to researchers, the use of basic extraction promotes the yield of heterogeneous feruloylated arabinoxylans with a molecular weight within the range of 2 kDa–2000 kDa under mild conditions, and with the introduction of a stronger base, with fractions of 100 kDa–2000 kDa and 2 kDa–7.6 kDa due to the deeper impact [53].

It is important to note here that in our case, the catholyte properties corresponded to a soft base judging by its effect on the arabinoxylan complex and arabinose and xylose small yields of 0.24% and 0.31%, respectively.

It is known that the application of high temperatures (140 ÷ 210) °C and microwaves facilitate depolymerization, branching and deesterification of cellulose complex compounds with the formation of colored products (furan compounds); therefore, lower temperatures are preferable [20].

Furan compound 5-hydroxymethylfurfural is a concomitant compound in sugar conversion reactions (caramelization and Maillard reaction). It indicates the thermal decomposition of sugars [54]. Gallic acid is reported to reduce the formation of 5-hydroxymethylfurfural, possibly preventing the oxidation of intermediate products of the Maillard reaction, while increasing the intensity of darkening, or it remains at a constant level [54]. According to the data obtained in our study, gallic acid is associated with thiol-containing amino acids whose sulfur group has antioxidant properties. Thus, it is possible that amino acids with thiol groups may have a synergistic effect in combination with phenolic acids toward the regulation of furan compound formation. On the other hand, the authors reported that furan can be formed in food products as a result of the thermal decomposition of not only carbohydrates but also the dissociation of amino acids [55]. However, not all amino acids can form furan compounds directly. Only serine and cysteine are transformed to aldehydes through thermal decomposition and then via aldol condensation form furan [56].

The pH of the medium is important for furan compound formation. Thus, the authors describe studies where the levels of furan formed at different pH levels and temperatures in glucose–glycine and fructose–glycine model systems differ [57]. At temperatures below 110 °C, the furan compound formation reaction may not take place at any levels of pH. At temperatures above 120 °C, the pH has a significant effect on furan formation: Acidic or alkaline pH values cause insignificant furan formation from glucose and/or fructose compared to a pH of 7.00. In this regard, a catholyte is a promising solvent.

The transformation of cysteine into furan compounds at different pH levels was investigated. It has been shown that an acidic pH contributes to the formation of increased amounts of sulfur-substituted furans, mercaptoketones and alkylfurans [58]. These data confirm the results of Table 2, Table 3 and Table 4 when the control values of the furan compound amount exceeded the experimental ones obtained on the basis of a catholyte with a pH of 9.3. The catholyte pH in the basic zone can also affect the conformational structure of sulfur-containing amino acids. It is shown that the geometry of the SH-SH bridge is a formed chemical adsorption unit, and with a pH below 5.5, the carboxyl group has a greater force during the interaction, while under alkaline conditions, on the contrary, the amino group acquires significance [59].

We obtained ambiguous data on the catholyte medium during the catechin extraction (Table 2, Table 3 and Table 4); the catechin content fluctuated compared to the control. Catechins are known to modify their configuration in slightly alkaline solutions, but in the presence of antioxidants (ascorbic or gallic acid), catechin transformation is inhibited [60]. Under catechin ultrasonic extraction conditions, the pH level and temperature of the process matter. It is claimed [61] that pH = 6.0 and 70 °C were the optimal conditions for the release of catechins compared to pH = 8.0 and 50 °C, which is in agreement with our study (Table 4) when a lower yield of catechins from the brewer’s spent grain was observed at a pH of 9.3 and 50 °C.

With regard to catechin extraction under excess pressure, a greater number of them were observed in the catholyte extracts obtained under a pressure of 0.5 atm or 110 °C.

Perhaps the explanation for this fact is that the catechin in the grain structure is associated with sugars that affect the polyphenol’s stability [62,63]. As a result of hydrolysis conditions, mineral components can be recovered from the brewer’s spent grain structure. These mineral components increase the catechins’ stability (for example, Cu^2+^ and Mn^2+^ ions) [64]. On the other hand, the structure of the phenolic compound molecule is significant. Thus, quercetin and catechin have the same number of hydroxyl groups in the same positions, but the former is characterized by a 2,3-double bond in the C-ring and the 4-oxo function [64], which makes quercetin more stable than catechin, which has a saturated heterocyclic ring in its structure.

Regarding the rutin content in the brewer’s spent grain, Yao et al. [65] report that it was in a bound state and was released after enzymatic treatment, and the composition of ethanol extract of the treated barley glumes included rutin (3.5 mg/100 g) and phenolic acids (vanillic acid (2.1 mg/100 g), ferulic acid (1.9 mg/100 g), coumaric acid (1.1), gallic acid (0.680), protocatechuic acid (0.3 mg/100 g) and p-coumaric acid (0.08 mg/100 g).

Thus, judging by the data of Table 2, Table 3 and Table 4, rutin, when extracted under excess pressure conditions in aqueous extracts, was optimally recovered under a pressure of 0.5 atm/110 °C, but in combination with a catholyte with minimal processing time that increased its content, ultrasound and catholyte treatment reduced the concentration in extracts proportionally to the increase in processing time. The increase in treatment time proved to have a positive effect on the rutin yield when extracted under conditions of atmospheric pressure and 50 °C in the presence of the catholyte. Thus, the main factors for rutin extraction under catholyte conditions are the limitations of the temperature and time of the extraction treatment. Under conditions of excess pressure of 0.5 ÷ 1 atm, a greater rutin yield is achieved when the time of the extraction treatment is 30 min. There are studies [66] in which the results show the stability of the grain rutin structure after heat treatment up to 150 °C and the degradation of the structure by breaking the C-C bond in the quercetin-3-rutinoside fragment at higher temperatures. It is possible that the conditions of prolonged (more than 30 min) treatment at a temperature of (110 ÷ 120) °C, which are created under excess pressure, facilitate the degradation of the rutin structure.

## 4. Materials and Methods

### 4.1. The Research Materials

Barley malt (Russia) sample was mixed with water in a ratio of 1:4 and mashed in an infusion way at the «Easy Drew» pilot brewery (Rybinsk, Russia). Mash was filtered, and brewer’s spent grain sample was washed in water several times, put into kraft bags and stored at a temperature of (0 ± 2) °C before this study.

### 4.2. Extraction Procedure

The brewer’s spent grain samples, obtained according to p. 4.1, were mixed with ECA-water catholyte pH 9.3 (experimental samples) in a ratio of 1:25 and processed: (a) at a temperature of 50 °C in thermal equipment for 60 and 120 min; (b) at over pressure (0.5–1.0 atm) for 30–120 min in autoclave and (c) in an ultrasonic bath at 50 °C for 30–120 min.

Control samples were also obtained; only distilled water was used as an extractant. After extraction, the samples were either filtered or centrifuged.

### 4.3. The Research Methods

#### 4.3.1. Chemicals

All reagents and standards were of analytical grade. Quercetin, rutin, catechin, arabinose, xylose, glucose, fructose, phenolic acids and aldehyds, 5-OH-methil furfurol, 5-methilfurfurol and furfurol standards were from Sigma-Aldrich with a purity of ≥99%. Potassium dihydrogen phosphate (KH_2_PO_4_), acetonitrile, acetic acid, orthophosphoric acid (H_3_PO_4_), ammonium dihydro-phosphate (NH_4_H_2_PO_4_), sodium monophosphate (Na_2_HPO_4_), sodium molibdate (Na_2_MO_4_), ninhydrin and potassium jodate (KJO_3_) were purchased from Galachem (Moscow, Russia).

Sulfuric acid, boric acid, hydrochloric acid (HCl), 5,5′-dithiobis [2-nitrobenzoic] acid, sodium bicarbonat (Na_2_CO_3_), phosphomolybdic acid and glycine were purchased from limited liability company “Reatorg” (Moscow, Russia).

Chemicals for determination β-glucan content were purchased in Megazyme Int. (Lansing, MI, USA).

Bidistilled prepared water was used in the determinations.

#### 4.3.2. Equipment

The following equipment was used to achieve the objectives of this scientific research: thermal equipment Carbolite Gero AX 60 (Carbolite Gero, Neuhausen, Germany), autoclave YXQ-LB-75SII (Boxun, Shanghai Boxun Industry and Commerce Co., Ltd., Shanghai, China) and ultrasonic equipment Skymen (40 kHz, 240 W) (Skymen, Guangming, Shenzhen, China).

#### 4.3.3. Determination of Nitrogen Compounds

To determine the common amount of soluble nitrogen, the Kjeldahl method (EBC Method 4.9.3) was used [67].

#### 4.3.4. Determination of Free Amino Acids (FAN) Amount

To determine the amount of free amino acid content, the ninhydrin manual method (EBC Method 8.10.1) was used [68].

#### 4.3.5. Determination of Low Molecular Weight Nitrogen

To determine the amount of low molecular weight nitrogen, 10 mL of sodium molybdate reagent (50% *w*/*w* water solution) was added to a 200 mL volumetric flask with aliquot of 125 mL sample. The samples were brought to the marked volume in the flask, and 10 mL of sulfuric acid was added. The sample was filtered, and the soluble nitrogen compound concentration was measured in filtrate (p. 4.2.2).

#### 4.3.6. Determination of Non-Protein Nitrogen

To determine non-protein nitrogen, the amount of amine nitrogen (p. 4.2.3) was subtracted from the amount of low molecular weight fraction soluble nitrogen (p. 4.2.4).

#### 4.3.7. Determination of Soluble Nitrogen with Thiol Groups Mass Concentration

We employed Elman’s method for determining the mass concentration of nitrogen with thiol groups. A total of 3 mL of a protein solution from 2 mL sample of 0.2 M phosphate buffer (pH 8) and 5 mL of distilled water (sample A) were added to a 20 mL test tube (sample A). Then, 10 mM Ellman’s reagent was prepared this way: 37 mg of 5,5′-dithiobis [2-nitrobenzoic] acid was dissolved in 10 mL of 0.1 M potassium phosphate buffer pH 7.0 and stirred. After that, 15 mg of sodium bicarbonate was added to the resulting solution and mixed again. Next, 3 mL of sample A was mixed with 0.02 mL of Ellman’s reagent, which was added with a micropipette. The sample optical density was measured on a spectrophotometer DR 3900 (HACH-LANGE, GmBH, Weinheim, Germany) at a wavelength of 412 nm after 3 min of exposure [69].

#### 4.3.8. Determination of Catechin Mass Concentration

The determination of catechin mass concentration was performed via high-performance liquid chromatography method with an “Agilent Technologies 1200” LC system (“Agilent Technologys”, Santa Clara, CA, USA) equipped with a diode array detector. HPLC equipment was fitted column Supelco C18 150 × 4.6 mm 5 μm (Thermo, Waltham, MA, USA) with wavelength 280 nm. The samples and all standard solutions were injected at a volume of 10 μL in a reversed-phase column at 25 °C. HPLC mobile phase was prepared as follows. Solution A: 50 mM NH_4_H_2_PO_4_ +1.0 mL of orthophosphoric acid dissolved in 900 mL of HPLC grade water, and the volume was made up to 1000 mL with water, and the solution was filtered through 0.45 μm membrane filter and degassed in a sonicator for 3 min. Solution B: acetonitrile. Mobile phase was run using gradient elution: at the time 1 min 5% B; at the time 10 min 15% B; at the time 10 to 45 min 40% B; at the time 45 to 55 min 98% B and at the time 55 to 60 min 5% B. The mobile phase flow rate was 1.2 mL/min, and the injection volume was 10 μL [70].

#### 4.3.9. Determination of Quercetin and Rutin Mass Concentration

The determination of the quercetin and rutin mass concentration was via high-performance liquid chromatography method with an “Agilent Technologies 1200” LC system (“Agilent Technologies”, Santa Clara, CA, USA) equipped with a diode array detector. HPLC equipment was fitted Luna 5 u C18 (2) 250 × 4.6 mm 5 μm (Phenomenex, Torrance, CA, USA) column with 290 nm wavelength. The samples and all standard solutions at a volume of 20 μL were injected into a reversed-phase column at 25 °C. The mobile phase was 2% acetic acid solution (A) and acetonitrile solution (B) with the ratio (A:B—70:30). The eluent flow rate was 1.5 mL/min [71].

#### 4.3.10. Determination of Phenolic and Furan Compound Mass Concentration

The determination of the phenol and furan compound mass concentration was via high-performance liquid chromatography method with an “Agilent Technologies 1200” LC system (“Agilent Technologies”, Santa Clara, CA, USA) equipped with a diode array detector. HPLC equipment was fitted with Phenomenex Hypersil ODS (C18) 250 × 4.6 mm 5 μm (Phenomenex, Torrance, CA, USA) column with 270 ÷ 330 nm wavelength. The samples and all standard solutions at a volume of 20 μL were injected into a reversed-phase column at 25 °C. The mobile phase was 0.5% phosphoric acid solution (A), acetonitrile solution (B) and 60 molar aqueous monopotassium phosphate solution (C) with the ratio (A:B:C—0.5:23:77). The eluent flow rate was 1.0 mL/min [72].

#### 4.3.11. Determination of the Mass Concentration of β-Glucan

To quantify the mass concentration of β-glucan, the standard fermentation method was used (8.13.1) [73].

#### 4.3.12. Determination of the Mass Concentration of Carbohydrates (Monomeric Sugars)

The determination of the carbohydrate (sugar) mass concentration was via high-performance liquid chromatography method with an “Agilent Technologies 1200” LC system (“Agilent Technologies”, Santa Clara, CA, USA) equipped with a refractive index detector. HPLC equipment was fitted with Luna 5u NH2 (2) 250 × 4.6 mm 5 μm (Phenomenex, Torrance, CA, USA) column. The samples and all standard solutions at a volume of 20 μL were injected into a reversed-phase column at 25 °C. The mobile phase was acetonitrile solution (A) and water solution (B) with the ratio (A:B—75:25). The eluent flow rate was 1.2 mL/min [74].

#### 4.3.13. Statistical Analysis

Statistical analysis was performed in five replicates. Descriptive statistics were performed, and values are expressed as mean ± standard deviation (SD). In this study, the Student–Fisher method was used; as a result of which, multivariate models of the correlation–regression dependence of the studied parameters were obtained. The reliability limit of the obtained data (*p* ≥ 0.95) was considered to assess various factors affecting the content of polyphenols in all studies; statistical data were processed by the statistics program (Microsoft Corporation, Redmond, WA, USA, 2006).

## 5. Conclusions

In this article, we have considered the effect of treatment with electrochemically activated water (catholyte with pH of 9.3) on brewer’s spent grain structure by evaluating the composition of the waste grain extracts under atmospheric and excess pressure, a temperature range of (50 ÷ 120) °C and a 40 kHz ultrasound. It was shown that treatment with a catholyte as a gentle basic agent allows for the efficient extraction of carbohydrate, nitrogen and monophenolic compounds under conditions of excess pressure, whereas sensitive flavonoids (catechins, quercetin and rutin) required less time for the extraction treatment and excess pressure to preserve their structure.

## Figures and Tables

**Table 1 molecules-28-04553-t001:** ECA-water usage in food industry.

Application Area	Application Effect	References
Microbiological	Catholyte showed low antimicrobial activity against *Candida albicans* (106 cells/mL) for 5 min; anolyte showed high activity. The preservation of the inhibitory properties of the anolyte for 2 months (in the dark at room temperature) and the catholyte—no more than 2 weeks—were shown.	[17]
Meat	Meat raw material’s biomodification with a high content of connective tissue with catholyte at pH 6.8–7.7 and temperature 50–55 °C without the introduction of chemical components with improved rheological properties of meat	[15]
Bakery	anolyte has an oxidizing effect on flour proteolytic enzymes; catholyte has a reducing effect; when processing flour from germinated wheat grains or with weak gluten, it is recommended to use an aqueous anolyte to strengthen the structure of gluten proteins when kneading dough from frozen flour or heavily dried grain—catholyte.	[13]
Brewing	Increase in proteolytic and amylolytic activity of cereal crops germinated using ECA solutions at a germination temperature of 15–20 °C	[18]

**Table 2 molecules-28-04553-t002:** Organic compounds content of brewer’s spent grain extracts.

Organic Compounds	Content in Extracts under Atmospheric Pressure and Treatment Time in the Medium (Reliability Limit *p* ˂ 0.05)			
	60 min		120 min	
	Control	Catholyte	Control	Catholyte
	carbohydrates, mg/L			
β-glucan	(38.8 ± 3.5) *	48.5 ± 4.4	46.6 ± 4.2	67.9 ± 6.1
	monosaccharide’s, g/L			
arabinose	nf **	nf	nf	nf
xylose	nf	0.015 ± 0.002	nf	nf
glucose	nf	0.511 ± 0.05	0.310 ± 0.03	nf
fructose	nf	0.172 ± 0.02	0.051 ± 0.005	0.110 ± 0.01
	nitrogen compounds, mg/L			
total soluble nitrogen	117.7 ± 4.7	117.7 ± 4.7	120.5 ± 5.0	126.1 ± 5.2
FAN	61.6 ± 2.5	70.0 ± 2.8	67.2 ± 2.5	81.2 ± 3.2
soluble nitrogen of low molecular weight fraction	61.6 ± 2.5	75.6 ± 3.0	61.6 ± 2.5	87.2 ± 3.2
non-protein nitrogen	nf	5.6 ± 0.2	nf	6.0 ± 0.2
nitrogen with thiol groups of low molecular weight fraction, μM/L	11.8 ± 0.6	22.4 ± 1.0	9.2 ± 0.5	13.2 ± 0.7
nitrogen with thiol groups and FAN ratio	1:5.2	1:3.1	1:7.3	1:6.2
	phenolic compounds, mg/L			
catechin	6.81 ± 0.60	8.66 ± 0.90	8.66 ± 0.9	10.87 ± 1.00
rutin	12.81 ± 1.00	13.82 ± 1.50	12.93 ± 1.5	29.85 ± 3.00
quercetine	0.25 ± 0.03	0.27 ± 0.03	0.28 ± 0.03	0.38 ± 0.04
gallic acid	1.65 ± 0.20	1.62 ± 0.20	1.66 ± 0.2	1.75 ± 0.20
vanillic acid	0.10 ± 0.01	0.08 ± 0.01	0.09 ± 0.01	0.08 ± 0.01
syringic acid	0.06 ± 0.006	0.07 ± 0.01	0.11 ± 0.01	0.11 ± 0.01
synaptic acid	nf	nf	nf	nf
vanillin	0.020 ± 0.002	0.044 ± 0.004	nf	nf
sirenic aldehyde	0.29 ± 0.03	0.26 ± 0.03	0.32 ± 0.03	0.32 ± 0.03
	furan compounds, mg/L			
5-OH-methylfurfurol	0.10 ± 0.01	0.09 ± 0.01	0.05 ± 0.01	0.05 ± 0.01
furfural	nf	nf	nf	nf
5-methylfurfurol	nf	nf	nf	nf

*—each value represents the mean of five independent experiments (±SD); **—lies outside the method scope.

**Table 3 molecules-28-04553-t003:** The organic compound content of brewer’s spent grain excess pressure extracts.

Organic Compounds	Content in Brewer’s Spent Grain Extracts at Excess Pressure and Treatment Time in the Medium at Pressure, Atm (Reliability Limit *p* ˂ 0.05)											
	30 min				60 min				120 min			
	Control		Catholyte		Control		Catholyte		Control		Catholyte	
	0.5	1	0.5	1	0.5	1	0.5	1	0.5	1	0.5	1
	carbohydrates											
β-glucan	(58.5 ± 5.3) *	48.2 ± 4.3	67.9 ± 6.1	56.3 ± 5.2	58.5 ± 5.3	48.2 ± 4.3	67.9 ± 6.1	77.6 ± 7.0	77.6 ± 7.0	67.9 ± 6.1	78.2 ± 7.0	85.3 ± 7.7
	monosaccharides, g/L											
arabinose	nf **	nf	nf	nf	nf	nf	nf	nf	nf	nf	nf	0.034 ± 0.003
xylose	nf	nf	nf	0.044 ± 0.004	0.034 ± 0.003	0.020 ± 0.002	0.022 ± 0.002	0.020 ± 0.002	0.020 ± 0.002	0.020 ± 0.002	0.043 ± 0.004	0.031 ± 0.003
glucose	nf	0.460 ± 0.05	nf	0.460 ± 0.05	0.800 ± 0.08	0.675 ± 0.070	0.814 ± 0.08	0.710 ± 0.070	0.725 ± 0.070	0.724 ± 0.070	0.682 ± 0.070	0.581 ± 0.060
fructose	0.160 ± 0.02	0.20 ± 0.02	nf	0.210 ± 0.02	0.230 ± 0.02	0.164 ± 0.020	0.216 ± 0.020	0.211 ± 0.020	0.159 ± 0.020	0.189 ± 0.020	0.207 ± 0.020	0.287 ± 0.030
	nitrogen compounds, mg/L											
total soluble nitrogen	131.7 ± 5.3	173.7 ± 7.0	135.7 ± 5.3	173.7 ± 7.0	201.7 ± 8.0	215.7 ± 8.6	215.7 ± 8.6	224.1 ± 9.0	187.7 ± 7.5	187.7 ± 7.5	221.3 ± 9.0	201.7 ± 8.0
FAN	58.8 ± 2.4	61.6 ± 2.4	75.6 ± 3.0	72.8 ± 3.0	58.8 ± 2.4	84.0 ± 3.2	64.4 ± 2.5	89.6 ± 3.2	89.6 ± 3.2	70.0 ± 2.8	89.6 ± 3.5	56.0 ± 2.2
soluble nitrogen of low molecular weight fraction	72.8 ± 3.6	67.2 ± 3.4	75.6 ± 3.6	75.6 ± 3.6	75.6 ± 3.6	84.0 ± 4.2	78.4 ± 3.6	86.8 ± 4.2	75.6 ± 3.6	81.2 ± 4.1	89.6 ± 4.3	70.0 ± 3.5
non-protein nitrogen	14.0 ± 0.5	56.0 ± 2.2	nf	28.0 ± 1.1	16.8 ± 0.7	nf	14.0 ± 0.5	nf	nf	11.2 ± 0.4	nf	14.0 ± 0.5
nitrogen with thiol groups of low molecular weight fraction, μM/L	23.7 ± 1.2	31.6 ± 1.6	19.2 ± 1.0	26.3 ± 1.3	7.1 ± 0.4	7.9 ± 0.4	27.4 ± 1.4	23.7 ± 1.2	6.6 ± 0.3	8.9 ± 0.4	26.3 ± 1.3	15.8 ± 0.8
nitrogen with thiol groups and FAN ratio	1:2.5	1:2	1:4	1:2.7	1:8	1:10	1:2.4	1:3.8	1:13.6	1:8	1:3.4	1:3.5
	phenolic compounds, mg/L											
catechin	1.24 ± 0.10	2.47 ± 0.20	6.19 ± 0.60	2.47 ± 0.20	8.66 ± 0.90	4.94 ± 0.50	4.94 ± 0.50	7.43 ± 0.70	10.52 ± 1.00	3.71 ± 0.40	6.81 ± 0.70	2.47 ± 0.20
rutin	10.55 ± 1.00	6.93 ± 0.40	25.29 ± 2.50	27.26 ± 2.70	8.77 ± 0.90	7.13 ± 0.70	10.45 ± 1.00	11.18 ± 1.10	3.79 ± 0.40	1.19 ± 0.10	1.61 ± 0.20	0.99 ± 0.10
quercetine	0.26 ± 0.03	0.28 ± 0.03	0.44 ± 0.04	0.66 ± 0.07	0.43 ± 0.04	0.88 ± 0.09	0.58 ± 0.06	0.50 ± 0.05	0.73 ± 0.07	0.88 ± 0.09	0.27 ± 0.03	0.64 ± 0.06
gallic acid	2.04 ± 0.20	1.53 ± 0.20	nf	0.48 ± 0.05	1.65 ± 0.20	1.78 ± 0.20	1.32 ± 0.10	1.54 ± 0.10	1.58 ± 0.10	1.69 ± 0.20	1.41 ± 0.10	nf
vanillic acid	0.09 ± 0.01	0.12 ± 0.01	0.09 ± 0.01	0.11 ± 0.01	0.18 ± 0.02	0.16 ± 0.02	0.13 ± 0.01	0.20 ± 0.02	0.29 ± 0.03	0.24 ± 0.02	0.18 ± 0.02	0.16 ± 0.02
syringic acid	0.06 ± 0.01	0.07 ± 0.01	0.08 ± 0.01	0.06 ± 0.01	0.06 ± 0.01	0.12 ± 0.01	0.13 ± 0.01	0.09 ± 0.01	0.19 ± 0.02	0.17 ± 0.02	0.15 ± 0.01	0.07 ± 0.01
synaptic acid	0.020 ± 0.002	0.050 ± 0.005	0.030 ± 0.003	0.38 ± 0.04	0.46 ± 0.05	0.61 ± 0.06	0.47 ± 0.05	0.89 ± 0.09	1.24 ± 0.10	1.01 ± 0.10	1.40 ± 0.01	1.69 ± 0.02
vanillin	0.19 ± 0.02	0.33 ± 0.03	0.05 ± 0.05	0.43 ± 0.04	0.40 ± 0.04	0.57 ± 0.06	0.50 ± 0.05	0.61 ± 0.06	0.83 ± 0.08	0.85 ± 0.08	1.12 ± 0.10	1.09 ± 0.10
sirenic aldehyde	nf	nf	nf	0.040 ± 0.004	0.06 ± 0.01	0.06 ± 0.01	0.17 ± 0.02	nf	0.08 ± 0.01	nf	0.040 ± 0.004	0.050 ± 0.005
	furan compounds, mg/L											
5-OH-methylfurfurol	0.040 ± 0.004	0.10 ± 0.01	0.06 ± 0.01	0.07 ± 0.01	0.17 ± 0.02	0.40 ± 0.04	0.08 ± 0.01	0.27 ± 0.03	1.14 ± 0.10	0.66 ± 0.07	0.57 ± 0.06	0.67 ± 0.07
furfural	nf	nf	nf	nf	0.040 ± 0.004	0.11 ± 0.01	nf	0.040 ± 0.004	0.19 ± 0.02	0.13 ± 0.01	0.07 ± 0.01	nf
5-methylfurfurol	0.010 ± 0.001	0.040 ± 0.004	nf	0.040 ± 0.004	0.07 ± 0.01	0.10 ± 0.01	0.06 ± 0.01	nf	0.050 ± 0.005	nf	nf	nf

*—each value represents the mean of five independent experiments (±SD); **—lies outside the method scope.

**Table 4 molecules-28-04553-t004:** The brewer’s spent grain extract organic compound content under ultrasound conditions.

Organic Components	40 kHz and Duration Treatment in Medium (Reliability Limit *p* ˂ 0.05)					
	30 min		60 min		120 min	
	Control	Catholyte	Control	Catholyte	Control	Catholyte
	carbohydrates, mg/L					
β-glucan	(38.8 ± 3.5) *	58.2 ± 5.2	59.4 ± 5.3	67.9 ± 6.1	59.4 ± 5.3	67.9 ± 6.1
	monosaccharides, g/L					
arabinose	nf **	nf	nf	nf	nf	0.037 ± 0.004
xylose	0.012 ± 0.001	nf	nf	nf	nf	0.032 ± 0.003
glucose	0.032 ± 0.003	nf	nf	nf	nf	nf
fructose	0.08 ± 0.01	nf	nf	nf	nf	nf
	nitrogen compounds, mg/L					
total soluble nitrogen	117.7 ± 4.7	134.5 ± 5.4	145.7 ± 5.8	165.3 ± 6.5	140.0 ± 5.6	162.5 ± 6.5
FAN	75.6 ± 3.0	89.6 ± 3.6	72.8 ± 2.9	78.4 ± 3.1	70.0 ± 2.8	81.2 ± 3.5
soluble nitrogen of low molecular weight fraction	58.8 ± 3.0	56.0 ± 3.0	61.6 ± 3.0	70.0 ± 3.5	61.6 ± 3.0	75.6 ± 3.6
nitrogen with thiol groups of low molecular weight fraction, μM/L	13.2 ± 0.5	23.7 ± 0.9	10.5 ± 0.4	63.2 ± 2.5	23.7 ± 0.9	55.3 ± 2.2
nitrogen with thiol groups and FAN ratio	1:5.7	1:3.8	1:7	1:2.4	1:3	1:1.5
	phenolic compounds, mg/L					
catechin	4.33 ± 0.40	0.62 ± 0.06	4.40 ± 0.40	0.62 ± 0.06	4.41 ± 0.04	2.47 ± 0.2
rutin	15.6 ± 1.5	6.8 ± 0.7	7.9 ± 0.8	5.5 ± 0.5	6.9 ± 0.7	4.8 ± 0.5
quercetine	0.36 ± 0.04	0.49 ± 0.05	0.35 ± 0.03	0.49 ± 0.05	0.27 ± 0.03	0.28 ± 0.03
gallic acid	nf	nf	nf	nf	nf	nf
vanillic acid	nf	nf	nf	nf	nf	nf
syringic acid	nf	nf	nf	nf	0.071	0.067
synaptic acid	nf	nf	nf	nf	nf	nf
vanillin	nf	nf	nf	nf	nf	nf
sirenic aldehyde	0.05 ± 0.01	0.08 ± 0.01	0.06 ± 0.01	0.06 ± 0.01	0.05 ± 0.01	0.021 ± 0.002
sinapic aldehyde	nf	nf	nf	nf	nf	nf
	furan compounds, mg/L					
5-OH-methylfurfurol	0.07 ± 0.01	0.06 ± 0.01	nf	nf	nf	nf
furfural	0.030 ± 0.003	0.025 ± 0.002	nf	nf	nf	nf
5-methylfurfurol	nf	nf	nf	nf	nf	nf

*—each value represents the mean of five independent experiments (±SD); **—lies outside the method scope.

**Table 5 molecules-28-04553-t005:** Paired correlation coefficients.

Brewer’s Spent Grain Processing Conditions	Paired Correlation Coefficients				
	Free Amino Nitrogen (FAN)	Amino Nitrogen with Thiol Groups (SH)	Gallic Acid (Gal)	Vanillic Acid (Van)	Sirengic Acid (Ser)
	5-hidroxymethilfurfural (5OHMF)				
Atmosferyc pressure	0.71	0.37	−0.38	0.20	−0.99
Excess pressure	−0.04	−0.40	0.01	0.65	0.27
	Furfural (F)				
Excess pressure	−0.59	−0.23	−0.08	−0.15	0.31
	5-methylfurfural (5MF)				
Excess pressure	−0.35	0.60	0.24	−0.39	0.05

**Table 6 molecules-28-04553-t006:** Partial correlation coefficients.

Brewer’s Spent Grain Processing Conditions	Partial Correlation Coefficients		
	5-Methylfurfural	Furfural	5-Methylfurfufal
Atmosferyc pressure	FAN-Van/5OHMF (−0.96) SH-Gal/5OHMF (−0.96) SH-Van/5OHMF(−0.95) Gal-Van/5OHMF (0.82) FAN-SH/5OHMF (0.76) R^2^ = 0.89	-	-
Excess pressure	Gal-Van/5OHMF (0.85) SH-Gal/5OHMF (0.83) Gal-Ser/5OHMF (0.73) SH-Ser/5OHMF (0.70) R^2^ = 1.0	FAN-Ser/F (0.80) Gal-Ser/F (0.78) SH-Gal/F (0.76) FAN-Gal/F (0.70) R^2^ = 1.0	FAN-SH/5MF (0.88) Gal-Van/5MF (0.83) SH-Gal/5MF (0.79) FAN-Gal/5MF (0.76) Gal-Ser/5MF (0.72) R^2^ = 1.0

**Table 7 molecules-28-04553-t007:** The brewer’s spent grain organic compounds’ optimum values extracted in the presence of catholyte.

Organic Compounds	Content in Catholyte Extracts under Conditions					
	Atmospheric Pressure		Excess Pressure		Ultrasound	
	Value	Exposure Conditions	Value	Exposure Conditions	Value	Exposure Conditions
	carbohydrates, mg/L					
β-glucan	67.9	2 h, 50 °C	85.3	1 atm, 2 h	67.9	2 h, 40 kHz
	monosaccharydes, g/L					
arabinose	nf	-	3.4·10^−2^	1 atm, 2 h	3.7·10^−2^	2 h, 40 kHz
xylose	nf	-	4.3·10^−2^	0.5 atm, 2 h	3.2·10^−2^	2 h, 40 kHz
glucose	51.1·10^−2^	1 h, 50 °C	81.4·10^−2^	0.5 atm, 1 h	nf	-
fructose	17.2·10^−2^	1 h, 50 °C	28.7·10^−2^	1 atm, 2 h	nf	-
	nitrogen compounds, mg/L					
total soluble nitrogen	126.1	2 h, 50 °C	224.1	1 atm, 1 h	165.3	1 h, 40 kHz
FAN	81.2	2 h, 50 °C	89.6	0.5 atm, 2 h	89.6	0.5 h, 40 kHz
soluble nitrogen of low molecular weight fraction	87.5	2 h, 50 °C	89.6	0.5 atm, 2 h	75.6	2 h, 40 kHz
non-protein nitrogen	6.0	2 h, 50 °C	28.0	1 atm, 0.5 h	nf	-
nitrogen with thiol groups of low molecular weight fraction, μM/L	22.4	1 h, 50 °C	27.4	0.5 atm, 1 h	63.2	1 h, 40 kHz
nitrogen with thiol groups and FAN ratio	1:6.2	2 h, 50 °C	1:3.8	1 atm, 1 h	1:3.8	0.5 h, 40 kHz
	phenolic compounds, mg/L					
catechin	10.87	2 h, 50 °C	7.43	1 atm, 1 h	2.47	2 h, 40 kHz
rutin	29.85	2 h, 50 °C	27.26	1 atm, 0.5 h	6.85	0.5 h, 40 kHz
quercetin	0.38	2 h, 50 °C	0.65	1 atm, 0.5 h	0.49	1 h, 40 kHz

## Data Availability

All data were presented in this article.

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
