# Peer review of "Study of Brewer’s Spent Grain Environmentally Friendly Processing Ways"

_molecules, 2023, doi:10.3390/molecules28114553_

Round 1
Reviewer 1 Report
The valorization and utilization of by-products in the food industry is an important and interesting topic. Characterization of by-products is important in order to determine their utility value. Various extraction techniques are used for by-products characterization, and recent research relies on environmentally friendly methods and techniques. This is an extensive and interesting research, but in my opinion the main flaw in this manuscript is related to the experimental part. Important corrections should be made in terms of a detailed explanation of how the extraction was performed. Especially, bearing in mind that a comparison of different techniques has been performed. It is necessary to describe in detail all experimental procedures and experimental conditions. Furthermore, in manuscript there is no explanation about how the control sample was prepared in relation to which the results of other extraction procedures were compared.
I could recommend certain corrections that would, I believe, enhance the paper quality:
In Tables 3-5: The authors used term: „reability limit‟. I recommend to use the term „significance level‟ since statistical confidence interval was set for p<0.05. Present the results with significant figures in relation to the absolute error or standard deviation, i.e. perform the correct rounding of numbers.
Lines 82-83: (Part 2.1)The title of the chapter needs to be changed. ECA-water catholyte does not influence the structure of brewer's spent grain, but the composition of obtained extract. This is a general suggestion for the titles of subsections 2.2. and 2.3.
Lines 95-97: In statement: „When the brewer’s spent grain is being processed for 1 h using catholyte, the ß- glucan content in the extracts increases by 25%, and when the treatment lasts for 2 h – by 45.7%.‟ comparing to what levels, or results?
Line 186: The authors refer to catechin extraction. In experimental part the HPLC procedure for catechin determination is given. On the other hand, in the results and discussion, they use the term catechin and the term catechins. Was only catechin determined or were some other catechin compounds analyzed, for example epigallocatechin gallate. It is necessary to clarify.
Significant corrections of the English language and grammar are needed.
Author Response
The authors are grateful for the detailed review of article. All comments were accepted and corrected.

Reviewer 2 Report
1. The main question addressed by the research is the effect of treatment with electrochemically activated water (catholyte) on the extraction of valuable compounds from brewer's spent grain.
2. The topic of the research is both original and relevant in the field. It explores the application of electrochemically activated water as a practical solution to extract and recover valuable compounds from waste materials like brewer's spent grain. This approach aligns with the principles of circular economy and sustainable resource utilization, making it a valuable contribution to the field.
3. In terms of methodology, the authors have provided sufficient detail to understand the experimental procedures and analytical techniques used.
There is a typo in line 582 – “[...] molecular weight nitrogen [...]”.
4. The conclusions are consistent with the evidence and arguments presented and support/address the main question of the research.
5. The references provided are appropriate.
6. While the tables provided in the manuscript present the data clearly, the inclusion of figures alongside the tables could enhance the visual representation of the results and improve the overall readability of the manuscript.
Overall, the manuscript presents valuable insights into the effects of electrochemically activated water on brewer's spent grain.
Author Response
The authors express their gratitude for reviewing the article materials
